# Assessing whether the best land is cultivated first: A quantile analysis

Thierry Brunelle[ID][1]*, David Makowski[2,3]

**1** CIRAD, UMR CIRED, Nogent-sur-Marne, France, **2** Centre International de Recherche sur l'Environnement et le Développement, Nogent-sur-Marne, France, **3** UMR 211, INRAE, AgroParisTech, University Paris-Saclay, Thiverval-Grignon, France

\* thierry.brunelle@cirad.fr

**Data Availability Statement:** All data files are available at the following URLs: - HYDE 3.2.1: https://easy.dans.knaw.nl/ui/datasets/id/easy-dataset:74467/tab/2 - Agricultural Suitability: https://zenodo.org/record/3748350#.XzpPjegzbIU - Market Accessibility: http://www.ivm.vu.nl/en/

## Abstract

Classical land rent theories imply that the best land is cultivated first. This principle forms the basis of many land-use studies, but empirical evidence remains limited, especially on a global scale. In this paper, we estimate the effects of agricultural suitability and market accessibility on the spatial allocation of cultivated areas at a 30 arc-min resolution in 15 world regions. Our results show that both determinants often have a significant positive effect on the cropland fraction, but with large variations in strength across regions. Based on a quantile analysis, we find that agricultural suitability is the dominant driver of cropland allocation in North America, Middle East and North Africa and Eastern Europe, whereas market accessibility shows a stronger effect in other regions, such as Western Africa. In some regions, such as South and Central America, both determinants have a limited effect on cropland fraction. Comparison of high versus low quantile regression coefficients shows that, in most regions, densely cropped areas are more sensitive to agricultural suitability and market accessibility than sparsely cropped areas.

## Introduction

Cultivated land covers around 1,500 million hectares (Mha), representing nearly 12% of the Earth's land area [1, 2]. Besides their key role in food, feed and bioenergy supplies, cultivated areas have major impacts on the environment, including climate change, water pollution, and biodiversity loss [3–7]. Agricultural projections anticipate that, by 2050, up to 330 Mha of land will be required at the global scale for food and feed production [8, 9]. On the other hand, climate mitigation scenarios stress the importance of freeing up land to regrow forest or to produce bioenergy crops [10]. Faced with this dilemma, optimizing the use of cultivated land represents a major challenge: increasing the supply of biomass for food and non-food purposes while limiting negative impacts on climate and biodiversity [11, 12].

To this end, we need a better understanding of the factors driving the spatial distribution of cropland. It is commonly accepted that the suitability of land for cultivation, which itself depends on climatic conditions during the growing season and soil characteristics (e.g. soil moisture, pH, slope and soil carbon content), influences the location of cultivated areas [13]. Market accessibility is also viewed as a key driver of the spatial distribution of cropland, as it is

Organisation/departments/spatial-analysis-
decision-support/Market_Influence_Data/index.
aspx.

**Funding:** This article benefited from the French
state aid managed by the ANR under the
"Investissements d'avenir" programme with the
reference ANR-16-CONV-0003.

**Competing interests:** Authos don't have
competing interests.

essential for trading in agricultural products and purchasing key inputs (e.g. seeds and fertilizers) [14].

These factors are captured in the economic concept of rent (surplus), which is the basis of economic theories of land allocation [15]. According to these theories, land is used in such a way as to maximize the rent generated by its use. Market accessibility and agricultural suitability have been recognized as key determinants of land rents since the 19th century in Ricardo's and von Thünen's classic theories [16, 17]. According to these theories, land is assumed to be cultivated gradually in descending order according to its quality and its distance from the market, to quote Ricardo: "The most fertile, and most favorably situated, land will be first cultivated" [16]. In this paper, we refer to the highest grades of land in terms of potential productivity, location suitability or both as "best land". Today, these theories are still directly applied in some land use assessments [18, 19]. Many global land use models are rooted in classical rent theories by allocating land according to a profit function that depends on the intrinsic qualities of land provided by vegetation models (usually in terms of climatic potential yields) [20–22] or based on index of agricultural suitability [23]. Land supply elasticities are also generally used to determine land conversion rates in a given location. In this case, the elasticity is estimated based on assumptions derived from rent theories [24].

Several empirical studies at the local scale have investigated responses to agricultural suitability and market accessibility [25]. Although strong relationships have been observed between cultivated area and agricultural suitability and market accessibility in several European regions [26, 27], some croplands in China were recently moved to less fertile areas in response to urbanization dynamics [28]. At the land system level, several assessments reported ambiguous effects of some spatial determinants on land use, like for example a negative effect of market accessibility on agricultural land use [25].

In this paper, we provide a comprehensive empirical analysis of the effects of agricultural suitability and market accessibility on the allocation of cultivated areas in 15 world regions (see region map in Fig 3). Our analysis is based on global datasets including suitability values and accessibility indices and cropland fractions on a 30 arc-minute grid. Agricultural suitability is proxied by an index synthetizing the climatic, pedologic and topographic properties of land [29]. Market accessibility is represented by an index reflecting the travel time using different types of infrastructure to medium (>50khab) and large (>750khab) cities and large maritime ports [14]. In this paper, we refer to the areas with the highest suitability and/or accessibility indices as "best land". Using these indices, we display cropland distributions over quartiles of agricultural suitability and market accessibility in each of the 15 regions. This representation has the advantage of displaying the quality of cropland allocation in relation to the region's potential in terms of agricultural suitability and market accessibility, since each region has its own specific conditions for cropland settlement. We then estimate the response of cultivated land to change in agricultural suitability and market accessibility, using quantile regression models [30] fitted to global datasets. This econometric approach was chosen here to make as few assumptions as possible about the distribution of data.

## Patterns of cropland allocation

The spatially explicit datasets that are currently available provide information on the cultivated fraction at the grid cell scale. However, we cannot directly infer from the formulation of rent theories their implications on this variable. To clarify this point, we show four contrasted patterns of cropland allocation on Fig 1. Four hypothetical regions are considered including four land classes covering an increasing gradient of land quality (Q1 = lowest quality, Q4 = highest quality). Here, the quality of land can correspond to either agricultural suitability or market

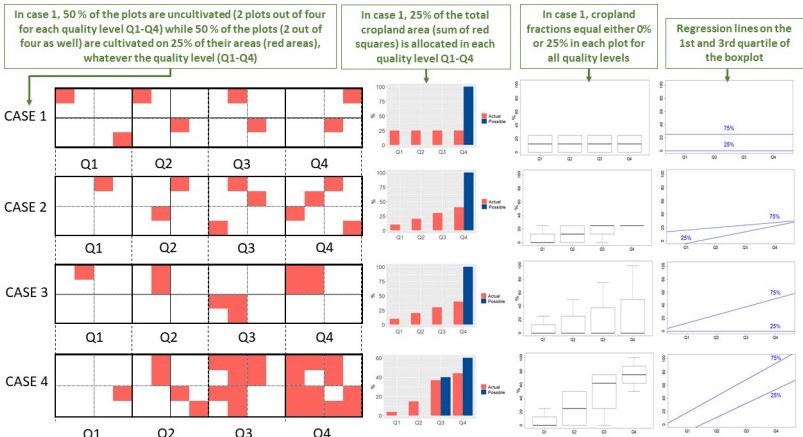

**Fig 1. Illustrative patterns of cropland allocation over a land quality gradient indicated by four land classes Q1, Q2, Q3, Q4 (Q1 = lowest quality, Q4 = highest quality) defined from the 1<sup>st</sup> quartile, median, and 3<sup>rd</sup> quartile of the considered quality index (suitability or accessibility).** Four hypothetical regions are distinguished, with low (case 1) to high (case 4) cultivated areas. Each region includes 16 parcels (white squares). A fraction ranging from 0 to 100% of each parcel can be cultivated (red squares in the graphics on the left). The histograms (center left) describe the corresponding cropland distributions expressed in % of agricultural land allocated to each land class (in red) in comparison to theoretically optimal land allocations (in blue). The boxplots (center right) describe the distributions of the cultivated fractions within land classes. The line charts (right) show the regression lines on the 25% least cultivated areas and the 75% most cultivated areas. Case 1 corresponds to a random and homogeneous distribution of the cultivated areas over the parcels. Case 4 illustrates a strong preference for an allocation of cultivated land in high quality parcels. Cases 2 and 3 are intermediate.

accessibility (see Method for details). Each of the four land classes is further divided into four parcels of homogenous quality on which cropland can be allocated and cover from 0 to 100% of the areas (red areas). The resulting distribution of cropland over land classes is shown on the histograms displayed in Fig 1 (center left). Depending on the proportion of cropland in the total land area, it may not be possible to allocate the entire area of cropland to the best land (i.e. in land class Q4). For this reason, the histograms compare the cropland distribution (in red) against a theoretical distribution (in blue) assuming that all cultivated land is allocated over the best parcels of land first. The boxplots (center right) describe the distribution of the cultivated fractions within land classes (i.e. the fraction of the parcel areas colored in red). The line charts (right) show the regression lines on the 25% least cultivated areas and the 75% most cultivated areas.

In the first hypothetical region (case 1), the cropland area (shown in red) is homogenously distributed over land classes, yielding a uniform distribution of cropland area over land types as well as a uniform distribution of crop fractions (25% in the four land classes). In the next two regions (cases 2 and 3), cropland is preferentially distributed to the highest classes of land quality either by increasing the number of cultivated plots (case 2) or by increasing the fraction of cultivated area within a given parcel (case 3). These two cases yield the same distribution of cropland area but with distinct distributions of crop fraction. In case 2, the crop fraction increases on the least densely cultivated parcels, making the lower range of the boxplot increase with land quality, while in case 3, the crop fraction increases on the most densely cultivated areas, making the higher range of the boxplot increase with land quality. In case 4, cropland is preferentially distributed on the best land class both by increasing the crop fraction and the number of cultivated plots. In this case, both the higher and lower ends of the boxplot increase with land quality. Note that, in case 4, we set the area of cropland of our hypothetical region to 40% of the total land area, which is larger than the area covered by the parcels of the

last land class (25%). Consequently, the theoretical distribution (in blue) spans the two last classes.

This schematic representation shows that the same distribution of cultivated areas can result from different spatial allocation strategies. Cropland can be preferentially distributed on the best land either by allocating crops to the least densely cultivated areas (better allocation) or by concentrating more crops on already densely cultivated areas (higher intensity) or both. These strategies lead to different graphical patterns in the boxplots of cultivated fractions. With the first strategy, we observe a steeper response at the lower end of the boxplot to land quality level whereas, with the second strategy, the response is steeper at the upper end of the boxplot (see right-hand charts on Fig 1).

Fig 1 summarizes the general approach adopted in this paper: we start from gridded data from which we derive crop distributions over land qualities (Q1-Q4). We then express these distributions as boxplots of cultivated fractions, and estimate the effect of land qualities for different quantiles of fractional crop coverage.

## Materials and methods

The analysis is carried out at global scale using datasets describing cropland fractions, agricultural suitability and market accessibility at the beginning of the 20th century at a 30 arc-min resolution. Data are scaled to values ranging between 0 and 1 where this is not already their native format. The fraction of cropland, infrastructure and other areas (including grassland and forest) in each grid cell comes from historical data based on HYDE version 3.2.1 [1] for the year 2017 without any distinction between crop types. HYDE 3.2.1 combines country statistics for different land use categories from FAO for the period 1960–2015, subnational levels statistics and spatially explicit depiction of land cover from the ESA Land Cover consortium maps for the year 2010. Data can be found at https://easy.dans.knaw.nl/ui/datasets/id/easy-dataset:74467/tab/2. Global agricultural suitability is measured using an index reflecting the climatic, soil and topographical conditions necessary to grow the 16 most important food and energy crops [29]. This index represents for each pixel the maximum suitability value across the 16 crop species. This index is strongly correlated with the Global Agro-Ecological Zones index [31]. As the Zabel's index is more recent, this index was chosen in our study. Data can be found at https://zenodo.org/record/3748350#.XzpPjegzbIU. Market accessibility of land is measured on the basis of the travel time using different types of infrastructure to medium-sized (>50khab) and large (>750khab) cities and large maritime ports integrated into a single index accounting for travel behavior [14]. Data can be found at http://www.ivm.vu.nl/en/Organisation/departments/spatial-analysis-decision-support/Market_Influence_Data/index.aspx. Maps of cropland fraction, agricultural suitability and market accessibility are shown on Fig 2 as well as zoomed-in maps for North America and Brazil on S15 Fig of S1 File. Results are aggregated for 15 agroclimatic regions (see Fig 3).

In order to assess the robustness of our conclusions as to the origin of the data, we compare our results to those obtained using cropland fraction from the Erb et al. land-use dataset [32]. We also use the market influence index, which incorporates population density data as well as national level per capita GDP values in addition to travel time to cities and ports [14]. Results, shown on S11-S14 Figs of S1 File, are consistent with those obtained with the default datasets.

### Building cropland distribution over suitability and accessibility quartiles

Quartiles of agricultural suitability and accessibility are calculated using the quantile function of R (v. 3.5.1). Quartiles are calculated for land where the suitability and accessibility index is not equal to 0 to avoid skewing the distributions in regions with large proportions of desert or

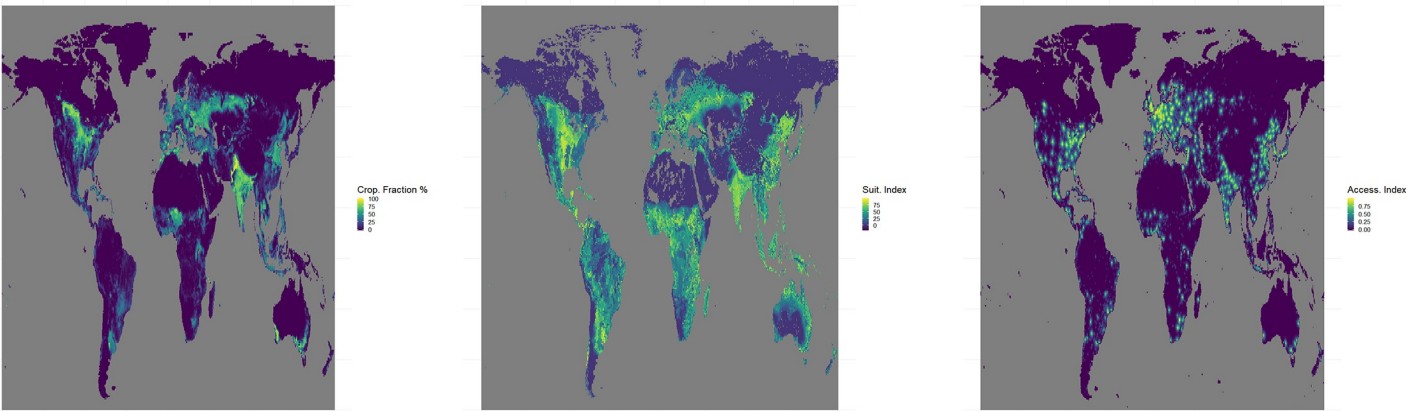

**Fig 2.** Cropland fraction in the year 2017 in percentage of grid cells from Goldewijk et al. (Panel A). Index of agricultural suitability (min = 0, max = 100) from Zabel et al. (Panel B). Index of market accessibility (min = 0, max = 100) from Verburg et al. (Panel C).

remote areas. The largest proportion of unsuitable land (index = 0) is found in Middle East and North Africa (14%) and Eastern Europe (9%), and the largest proportion of inaccessible land (index = 0) is found in Oceania and South-Eastern Asia (23%) and Southern Africa (11%).

Due to the relatively low resolution of the maps and the granularity of the indices (in terms of decimal place), the quartiles obtained do not exactly yield a homogeneous distribution of 25% land by quartiles. This is particularly the case for the suitability index, whose granularity

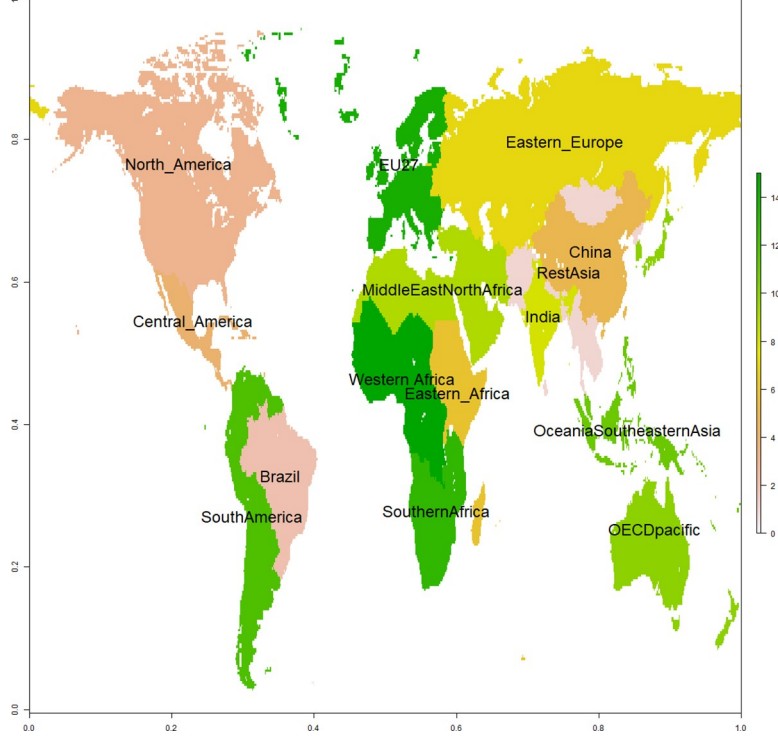

**Fig 3. Map of the 15 world regions.**

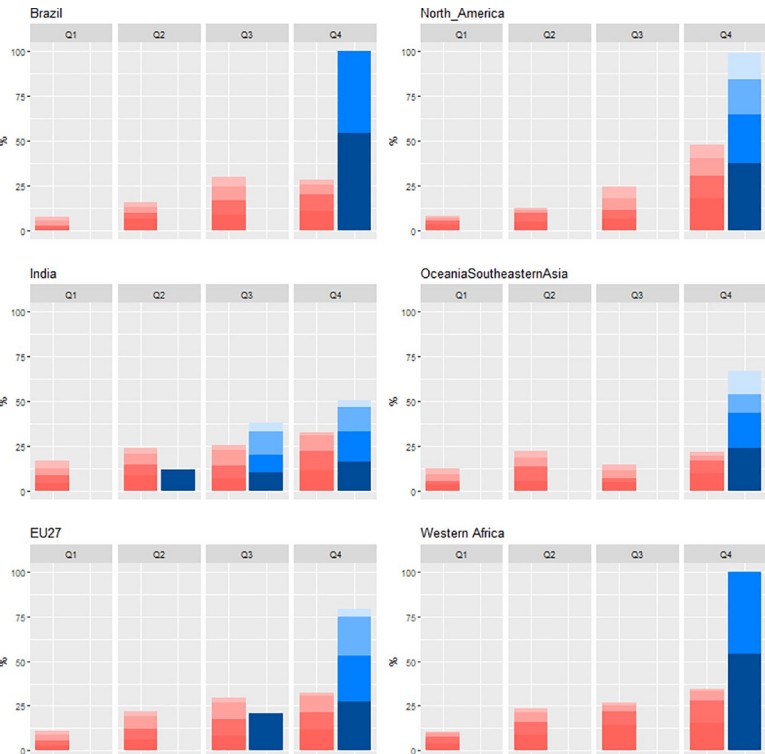

**Fig 4. Percentages of cropland area allocated to four different classes (Q1, Q2, Q3, Q4) of suitability (red) and theoretically possible distribution (blue) for six regions.** Shade levels indicate land classes of accessibility, the darkest colors showing the most accessible land. The four land suitability classes were defined from the 1st quartile, median, and 3rd quartile of the considered suitability index.

is limited to two digits after the decimal point (compared to seven for the accessibility index). The quartiles of agricultural suitability obtained using the quantile function default setting were compared to quartiles empirically calculated by iterative processes. The latter quartiles were used when they provided a more uniform distribution of 25% per quartile. The resulting distribution of total land area over quartiles of suitability and accessibility is provided in S1 and S2 Tables of S1 File.

The theoretical distribution of land shown in blue bars in Figs 1 and 4 is calculated by allocating an area equivalent to the cultivated area in a given region to the four land categories defined from the quartiles of suitability and accessibility. All types of land are included in the calculation–cropland, pasture, forest or wilderness areas–with the exception of areas occupied by infrastructure.

## Quantile regression

We use quantile regression to measure the effect of agricultural suitability and accessibility on cropland fraction [30]. This semi-parametric approach provides a more detailed picture than classical least square regression methods, as it focuses on the entire conditional distribution of the dependent variable, not only on its mean. It allows us to assess how cropland fractions are distributed over different types of land characterized by different quantiles (high quantiles correspond to more intensively cultivated land than low quantiles). Successively, for two quantile levels $\tau$ (1st and 3rd quartiles), we estimate the coefficients $\beta_0$, $\beta_1$, $\beta_2$, $\beta_3$ of the following

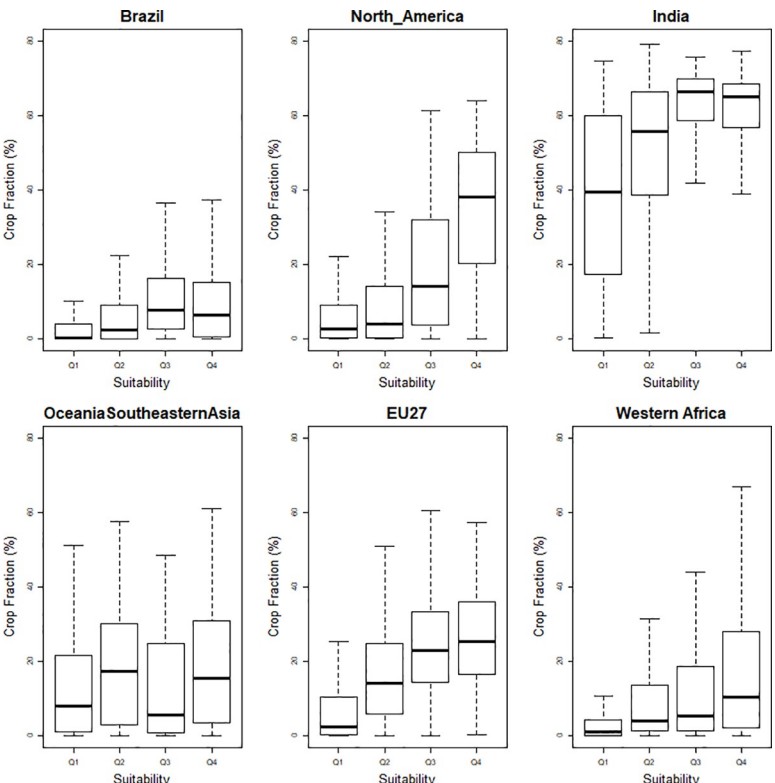

**Fig 5. Distribution of crop fractions (%) in areas of 30 arc-minute grid cells for four classes of suitability (Q1, Q2, Q3, Q4) in six regions.** The four land suitability classes were defined from the 1st quartile, median, and 3rd quartile of the considered suitability index.

quantile regression model:

$$Q_\tau(\text{CropFraction}) = \beta_0(\tau) + \beta_1(\tau) * \text{Suitability} + \beta_2(\tau) * \text{Accessibility} + \beta_3(\tau) * \text{Suitability} * \text{Accessibility}$$

The regression coefficients were estimated using the procedure described by Koenker [35] with the rq function of the R quantreg package.

Standard errors are obtained by bootstrap methods with 1,000 replications. The quality of fit of the quantile regression models is assessed using a pseudo R2 noted R1 for each specified quantile (here the first and third quartiles). This quality of fit criterion (in the range 0–1) was specifically designed for regression quantile models [33] and is expressed as a weighted sum of the values of the residues of the fitted quantile regression. R1 is a natural analog to the R2 for quantile regression and measures the local quality of fit at each fitted quantile.

Results for the 1st and 3rd quartiles are shown on Fig 6. Additionally, quantile regression models were fitted for each of the 15 relevant regions at a 5-percentile interval between the 10th and 95th percentiles. Results for all quantiles are shown on S5-S7 Figs of S1 File.

## Heteroscedasticity, multicollinearity and spatial autocorrelation

By fitting linear regressions to different conditional quantiles of the range of a response variable, quantile regression overcomes the problem of heterogeneity of variance [34] and is thus well suited in the presence of heteroscedastic error (the Breusch-Pagan test fitted to a linear regression model is highly significant).

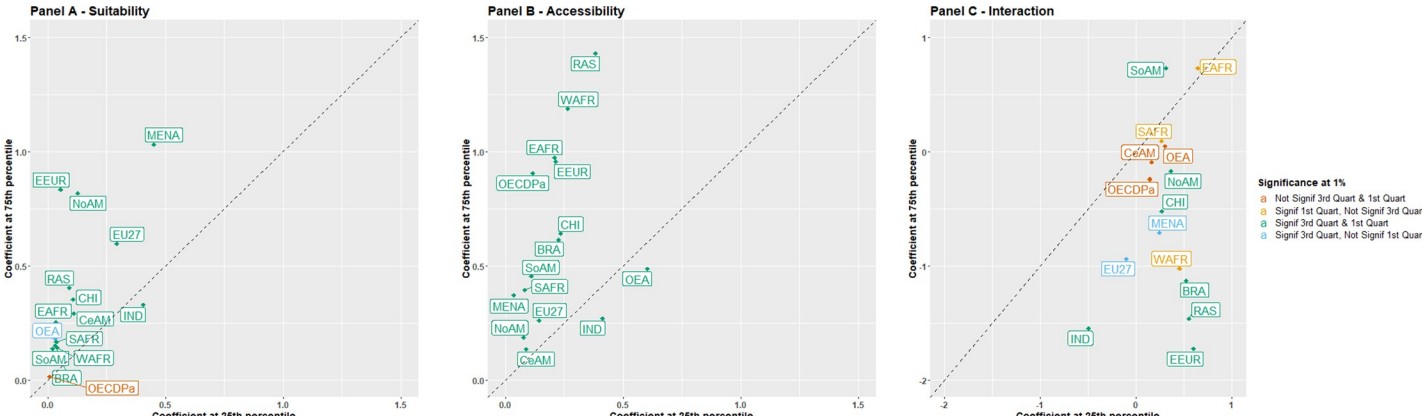

**Fig 6.** Estimated effects of agricultural suitability (Panel A), market accessibility (Panel B), and the interaction between suitability and accessibility (Panel C) for the first and third quartiles of cropland fractions in 15 regions. Estimates were produced by quantile regression. The first and third quartiles represent the 25% least- and most-cultivated land. Colors indicate the levels of statistical significance for each region. RAS: RestAsia, BRA: Brazil, NoAM: North America, CeAM:Central America, CHI: China, EAFR: Eastern Africa, EEUR:Eastern Europe, IND: India, MENA: Middle East and North Africa, OECDPa: OECD pacific, OEA: Oceania Southeastern Asia, SoAM: South America, SAFR:Southern Africa, EU27: European Union, WAFR: Western Africa.

To test for possible multicollinearity issues, we calculate the Spearman coefficient of correlation between independent variables (see S3 Table of S1 File). The resulting values are in most cases below 0.6. Regression coefficients have been estimated both through univariate and multivariate regressions: we did not find any noticeable difference in the values of the coefficients except for North America. In this region, we obtained a value for the accessibility coefficient of the multivariate regression about four times lower than that of the univariate regression. As there is only a moderate correlation between the independent variables in North America (Spearman = 0.49), we decided to use the value of the multivariate regression coefficient because of its lower Bayesian Information Criterion (BIC) values.

The Moran test for spatial auto-correlation confirms that both our dependent and independent variables are spatially autocorrelated. Thus, the spatial autocorrelation of our response variable (crop fraction) is probably caused by our autocorrelated predictors (suitability and accessibility). In this case, it is not relevant to remove this effect from our predictors because our objective is precisely to estimate the effects of suitability and accessibility on the crop fraction. However, in order to avoid any unintended effect resulting from a residual autocorrelation, all p-values and confidence intervals of our estimates were computed using a non-parametric bootstrap procedure with 1,000 replications. The corresponding confidence intervals are shown in S8-S10 Figs of S1 File and do not indicate any spurious effect.

## Results

### Cropland is preferentially distributed in areas with high levels of agricultural suitability and market accessibility in most regions

The cropland distributions over classes of agricultural suitability for a selection of world regions is shown in Fig 4. The four land classes are defined from the 1[st] quartile, median, and 3[rd] quartile of the considered suitability index. The distributions in all of the 15 regions studied, as well as distributions over classes of land accessibility, are provided in the Supplementary Information (see S1 and S2 Figs of S1 File). As in Fig 1, we compare the actual cropland distribution (in red) against a theoretical distribution (in blue) assuming that all cultivated land is distributed over the class of the most suitable land unoccupied by infrastructure (see Method).

It should be noted that the theoretical distribution does not necessarily represent an optimum to be achieved and is simply intended to show possible limitations in land availability.

The shape of the cropland distributions reveals that cultivated areas are in most cases allocated preferentially to the most suitable land, although there are substantial variations between regions. With 50% of its cropland area located on the most suitable class of land and only 8% on the least suitable land class, North America is the only region where the difference in cultivated land allocation between the lower and upper classes (noted Q4-Q1 in the following, for convenience) is greater than 30 percentage points. This Q4-Q1 difference is lower in India (16 percentage points), EU27 and Brazil (21 percentage points in each case). In India and EU27, the total cropland area is such that it is not possible to allocate all cropland beyond the 3rd (in EU27) or the 2nd quartiles (in India) of agricultural suitability. In Brazil, on the other hand, all crops could theoretically be located beyond the 3rd quartile, also on land with relatively good accessibility (see blue bars on Fig 4). Out of the 15 regions studied, 10 have a Q4-Q1 difference between 20 and 30 percentage points. The lowest Q4-Q1 differences are found in Oceania and South-Eastern Asia and Central America where the cropland distributions are almost uniform (see Fig 4 and S1 Fig of S1 File).

Compared to agricultural suitability, the concentration of crops on the most accessible land is greater: out of the 15 regions studied, seven have more than 40% cropland above the third class (see S2 Fig of S1 File), a value found only in North America with respect to agricultural suitability. The largest Q4-Q1 differences in terms of land accessibility are found in Middle East and North Africa, Rest of Asia, Eastern Europe and Western Africa, and the lowest in Oceania and South-Eastern Asia and OECD Pacific.

The crop fraction (%) in areas of 30 arc-minute grid cells are shown in Fig 5 for the same selection of regions and in S3 Fig of S1 File for the whole set of regions. The crop fraction tends to increase with agricultural suitability in most regions and the trends are consistent with the cropland distributions shown in Fig 4. The response is particularly strong in North America but is much weaker in Oceania and South-Eastern Asia. Fig 5 shows that, in most regions, the increase in crop fraction is stronger at the upper end of the boxplots than at the lower ends. India is a notable exception as the effect of agricultural suitability on the crop fraction is stronger in the least cultivated areas. This can be explained by the particularly dense crop cover in India, which implies that the effect of suitability is mainly obtained through a better allocation in the least cultivated areas rather than an increased intensity in the most cultivated areas. A similar upward trend was observed for crop fraction with respect to the accessibility index, with a steeper increase in the most cultivated areas in all regions except India and Oceania and South-Eastern Asia (see S4 Fig of S1 File).

## Quantification of the effects of agricultural suitability and market accessibility on cropland fraction

Quantile regression models were fitted for each of the 15 relevant regions to estimate the effects of a one-unit increase in suitability and accessibility on the cropland fraction. For each region, two models were fitted separately: one for the 25% most cultivated land, i.e. areas with crop fraction above the third quartile of the grid cells in our dataset, representing densely cropped areas; and one for the 25% least cultivated areas, i.e. areas with crop fractions lower than the first quartile of the grid cells, representing sparsely cropped areas (see details in Method). The estimated coefficients obtained for these two quantiles are plotted in Fig 6. Their comparison makes it possible to analyze the response of the fraction of cultivated land to a one-unit increase in suitability and accessibility, taking into account the intensity of agricultural land-use.

The results of the quantile regression on the 25% most cultivated land (3rd quartile) show that increasing suitability has a positive effect on the fraction of cultivated land for all regions (Fig 6A). This effect is always significant (p<0.01) with one exception in OECD Pacific. The suitability effect is higher than 0.5 in four regions, namely North America, EU27, Eastern Europe and Middle East and North Africa. According to these estimates, a 10% increase in suitability would increase the cropland fraction by more than 5% in these regions. However, in Eastern Europe, India, Rest of Asia and Brazil, this positive effect can be partially offset by a strong negative and significant interaction between suitability and accessibility (Fig 6C).

For the 25% least cultivated land (1st quartile), the estimated effects of suitability are almost always significantly above zero (p<0.01) with only two exceptions: Oceania and South-Eastern Asia and OECD Pacific (Fig 6A). The estimates are systematically lower on the 1st quartile, except in India which is the only region where the estimated coefficient is higher than 0.4. Thus, the response to agricultural suitability is achieved in most regions through a higher intensity in already densely cultivated areas (Case 3 in Fig 1) rather than through a better allocation of cropland in the least densely cropped areas (Case 2).

With respect to the accessibility index, the estimated effects of a one-unit increase in accessibility on cropland fractions in the 25% most cultivated land are higher than 0.75 in five regions: Rest of Asia, Western Africa, Eastern Europe, OECD Pacific and Eastern Africa (Fig 6B). Estimates are significantly higher than zero for all regions (p<0.01). As shown for suitability, the estimated coefficients are lower when considering the 25% least cultivated land, except in India and Oceania and South-Eastern Asia, which is consistent with the boxplots of crop fractions (see S4 Fig of S1 File).

Coefficients of interaction are significant at both the 1st and 3rd quartiles in only seven regions (Fig 6C). They are positive in most regions at the 1st quartile of crop fractions and negative in most cases at the 3rd quartile. Thus, in the most densely cropped areas, the effect of one of the two explicative variables becomes less important as the value of the other variable increases, while it becomes more important in areas with low crop density. The interaction between suitability and accessibility is particularly acute in Eastern Europe, Rest of Asia, India and Brazil.

## Ricardian and Von Thünen paths of cropland allocation

To assess the relative importance of suitability over accessibility, we use a standard criterion, called R1, frequently used with quantile regressions [33] to measure the quality of the model at a given quantile (see Method). High R1 values indicate a better explanatory power of the estimated model. This criterion is close to zero when the response is nearly flat and close to 1 when the quality of fit of the model is almost perfect for the relevant quantile. Here, we calculate R1 at the 1st and 3rd quartiles for two univariate models with agricultural suitability and market accessibility respectively as independent variables. The higher of the two R1 values in the two quartiles is reported in Fig 7.

R1 values are higher at the third quartile in all regions, except India, EU27, and Oceania and South-Eastern Asia (for accessibility only in the last case). The best fits are found in North America, Middle East and North Africa, Eastern Europe and China for both the suitability and accessibility indices. In these regions, the R1 values are equal to 0.3 or higher, meaning that the fit is improved by more than 30% compared to a model with the intercept only. R1 values of less than 0.2 confirm that agricultural suitability and market accessibility have a small effect on cropland fractions in South and Central America, OECD Pacific, Oceania and South-Eastern Asia and Southern Africa.

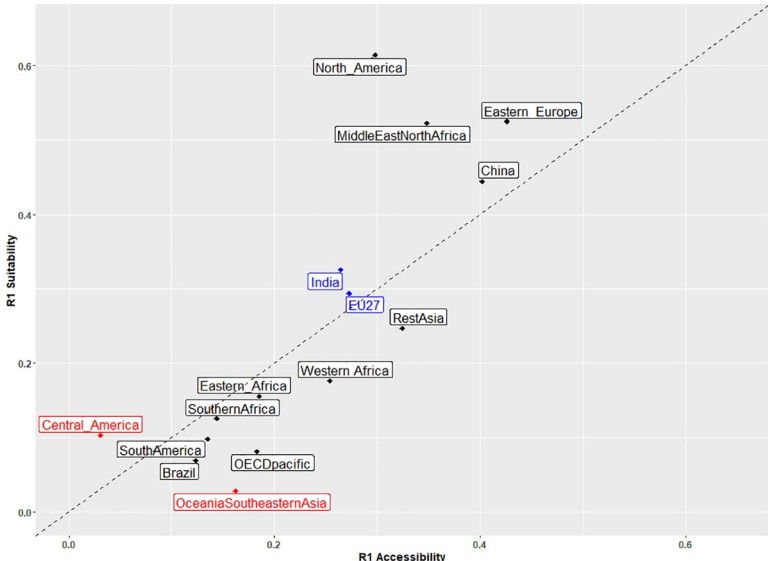

**Fig 7. R1 values of quantile regression of two univariate models linking crop fraction to agricultural suitability (y-axis) and market accessibility (x-axis).** The higher of the R1 values in the first and third quartiles of crop fraction distribution is reported. Values are computed at the third quartile for the regions labelled in black, at the first quartile for the regions labelled in blue, and at the third quartile regarding suitability (suit.) and the first quartile regarding accessibility (access.) for the region labelled in red.

The 1:1 line shown in Fig 7 separates regions where the highest degree of explanation is provided by agricultural suitability from those where it is provided by market accessibility. Referring to the underlying economic theories, the former regions can be labelled as Ricardian regions and the latter as Von Thünen regions. A number of regions–India, South America, Eastern Africa and China–lie near the 1:1 frontier, indicating a certain balance between accessibility and suitability criteria. At the top right of the R1 plot, North America shows a Ricardian-oriented allocation, with crop allocation highly concentrated in the highly suitable areas of the Corn Belt. At the bottom right, Rest of Asia and Western Africa, where crops are mostly located close to major rivers (Niger and Mekong) characterized by high levels of accessibility, can be described as Von Thünen regions.

The prevailing influence of one variable over the other might be related to the relative level of heterogeneity found in suitability and accessibility. Thus, if suitability levels are more heterogenous than accessibility, one might expect a stronger effect of suitability on cropland fractions. Conversely, in regions where agricultural suitability is relatively homogeneous, as for example in Brazil (see S15 Fig of S1 File), there will be little benefit in improving land allocation on this criterion. To test this assumption, we analyzed the relationship between the ratio of the estimated effects of suitability vs accessibility and the ratio of coefficients of variation (relative standard deviation) of suitability vs accessibility (see S4 Table of S1 File). Results show a significant relationship (p value < 0.01 and adjusted R-square = 0.39), thus confirming our assumption.

## Discussion

Our results show that the validity of the assumption that the best land is used first–arising from classical land rent theories–varies across the regions of the world. In most regions, agricultural suitability and market accessibility have a significant positive effect on the fractional

crop coverage, but their influence may be limited in some cases such as in Central and South America. Moreover, the prevailing driver of cropland allocation differs between regions. Agricultural suitability and market accessibility have a similar influence on crop fractions in several regions, in particular in China, India, and EU27. However, in North America, Middle East and North Africa and Eastern Europe, agricultural suitability appears to be a stronger driver of cropland allocation than accessibility.

Comparison of high vs. low quantile regression coefficients shows that, in most regions, cropland systems with high fractional crop coverage are more responsive to higher land grades than extensive or mosaic cropland systems characterized by sparser cultivated areas. This suggests that large-scale commercial farms are more likely to use the best land than smallholders and mixed crop-livestock agriculture. India and Oceania and South-Eastern Asia are two notable exceptions that can be explained by a higher proportion of smallholders engaged in commercial farming, especially in oil palm and rice production [35–37].

It is noteworthy that the best fits are found in regions with a long history of intensive agricultural settlement—North America, Middle East, Eastern Europe, China, India and European Union—while the poorest fits are found in regions where the agricultural frontier remains active —especially Brazil and Oceania and South Eastern Asia. This finding is consistent with the hypothesis of a process of gradual optimization of land allocation through learning described by Mather and Needle in the forest transition theory [38]. The United States is a "textbook" case of gradual optimization of land allocation. In the US, crop allocation were originally cultivated in relatively unsuitable land on the East Coast and in the Appalachian foothills [15, 39, 40]. The reduction in transport costs resulting from the development of the railways, combined with the reduction in transatlantic freight rates, made it possible to cultivate the fertile lands of the Midwest and export part of the agricultural production to Western Europe [39]. Our results are also consistent with the findings of several authors regarding the presence of increasing marginal returns in areas of agricultural expansion [41, 42] and support Di Tella's abnormal rent theory distinguishing between frontiers in equilibrium, where the price equals the average cost (zero profit) with a possibility of differential rent formation as one moves away from the frontier, and frontiers in disequilibrium with a possibility of positive profit and increasing returns [43].

This paper provides a number of insights that could help to improve the efficiency of land-use allocation and limit the pressure of agricultural activities on natural areas. Our results highlight substantial potential for improving the allocation of cropland in some major agricultural regions, particularly in South America and West Africa, where large amounts of the most suitable and accessible land are used for other purposes than crop production. Our results also suggest that the relative importance of agricultural suitability to market accessibility as a driver of cropland allocation in a given region is related to the relative level of variability of each determinant in that region. Thus, reducing the variability of market accessibility conditions through better transport infrastructure may foster the effect of agricultural suitability on cropland allocation and allow for more efficient use of the agronomic potential in a given region.

Coordination between public institutions and the private sector will be key to improving land-use patterns through, for example, the dissemination of information, fiscal incentives and facilitated provision of production factors [44]. Most importantly, promoting more equal access to land is essential to enable all types of agriculture, including smallholdings, to use the most suitable and accessible land. The environmental impact of such reorientation should also be taken into account in order to ensure that it benefits the conservation of natural areas.

Making land allocation more efficient also implies that it is possible to substitute different types of land uses, such as croplands, pastures and forest areas. However, the level of substitutability between different types of land uses depends on the interplay of many parameters, including cultural factors, the type of actors involved and local agricultural and environmental regulations

[44–46]. For example, important complementarities between land use types (e.g., forest vs. crop-land) in mosaic system can impede the transition towards more efficient cropland allocation.

Better representation of land-use changes is essential to decision-making. Here, we show that classical rent theories, which are still influential in land-use studies, cannot be applied independently of the regional context. They need to be used as contextualized generalizations rather than as "grand theories" [47] to account for the varying effects of spatial determinants of land allocation in the different regions of the world. Finally, it is important to emphasize that in many parts of the world market accessibility, which is sometimes overlooked in land-use studies, is a more important driver of cropland allocation than agricultural suitability.

There are several limitations to our work. First, we use a static dataset that does not provide an analysis of land-use transitions. This implies that our analysis spans the entire history of agricultural settlement in a given region, and may not detect recent changes in the spatial allo-cation of crops. Moreover, we consider total cultivated area without distinguishing the differ-ent crop types and their associated agronomic constraints. This could be a source of inaccuracy in certain regions, for example Oceania and South-Eastern Asia where the domi-nant crop is oil palm for which suitability is mainly dependent upon regular rainfall. In this study, we assume that the actual crop mix is close to the optimal one (i.e., the crop mix show-ing the highest suitability in a given grid cell). This seems a reasonable assumption because farmers generally tend to grow the species that are best adapted to their environment, i.e., to local climatic, soil and topographical conditions. In some cases, humans have nevertheless been able to shape their environment, through irrigation, drainage or terracing, to make it more suitable for agriculture. This may explain deviations from a perfectly efficient distribu-tion (i.e., where all crops would be on the best land). Also, climate change may make the choice of crops more complex, leading possibly to larger discrepancies between the actual and optimal crop mix. Finally, this analysis does not account for cropping intensity (i.e., the fraction of the cultivated area that is harvested). In doing so, we cannot conclude about Ester Boserup's cri-tique of rent theories. This critique states that rent theories are based on an "oversimplified conception" of the agricultural system distinguishing between cultivated and uncultivated land, while landscapes are actually shaped by a continuum of land types that differ in their fre-quency of cropping [48]. The two views may be however not entirely divergent from a land use perspective. One can certainly think that there is a link between density of cultivated areas and cropping intensity. Low densities may be a signal of long fallows, while high densities are usually associated with annual or multiple cropping. This hypothesis, which remains to be confirmed, would be a way of reconciling the two sides.

## Supporting information

**S1 File.**
(DOCX)

## Acknowledgments

Authors thank Tristan Lecotty, Patrice Dumas, Tamara Ben Ari and two anonymous reviewers for their valuable comments on earlier drafts of this article. This article benefited from the French state aid managed by the ANR under the "Investissements d'avenir" programme with the reference ANR-16-CONV-0003.

## Author Contributions

**Conceptualization:** Thierry Brunelle, David Makowski.

**Formal analysis:** Thierry Brunelle, David Makowski.

**Writing – original draft:** Thierry Brunelle, David Makowski.

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
