## [Decision Letter · Decision Letter 0]

27 Jul 2020

PONE-D-20-17711

Assessing whether the best land is cultivated first: A quantile analysis

PLOS ONE

Dear Dr. BRUNELLE,

Thank you for submitting your manuscript to PLOS ONE. After careful consideration, we feel that it has merit but does not fully meet PLOS ONE’s publication criteria as it currently stands. Therefore, we invite you to submit a revised version of the manuscript that addresses the points raised during the review process.

Reviewers have raised concerns about the data used and the need to provide a bit more description and justification for their use. There is also the need to improve the explanation for the selection and composition of the agricultural suitability index, and a justification for its choice over others. The discussion section will need a bit more effort to enable readers better appreciate the implications of the study as well as its limitations. Please, kindly review the figures in line with suggestions made by the reviewers.

We look forward to receiving your revised manuscript.

Kind regards,

Gerald Forkuor

Academic Editor

PLOS ONE

Journal Requirements:

Reviewers' comments:

Reviewer's Responses to Questions

**Comments to the Author**

1. Is the manuscript technically sound, and do the data support the conclusions?

Reviewer #1: Yes

Reviewer #2: Partly

2. Has the statistical analysis been performed appropriately and rigorously? 

Reviewer #1: I Don't Know

Reviewer #2: Yes

3. Have the authors made all data underlying the findings in their manuscript fully available?

Reviewer #1: Yes

Reviewer #2: No

4. Is the manuscript presented in an intelligible fashion and written in standard English?

Reviewer #1: Yes

Reviewer #2: Yes

5. Review Comments to the Author

Reviewer #1: PLOS One review of “Assessing whether the best land is cultivated first: A quantile analysis”

This paper explores the question of whether the best (i.e. most suitable and accessible) land is cultivated first, using quantile regressions on cropland fraction, agricultural suitability and accessibility at the global scale. The paper provides a very valuable and well-articulated contribution that I believe will be of interest to people studying issues related to land use. I therefore think that it would be a good paper for this journal if it addresses a few concerns. I am describing these below. I should note that I am not familiar with quantile regression as a method, and as such, my ability to comment on that aspect of the manuscript is limited, though the methods do seem sound.

Choice of data: In general I think the paper could better justify and describe the data, including how it is produced and what resolution it’s at – if space restrictions make it impossible in the paper, then a brief description in the SI would be fine. Specifically:

- I am not entirely clear why the authors chose to use the HYDE database as a source for cropland areas, even though the fact that they come to similar results when using another dataset (Erb 2007) is reassuring. HYDE’s purpose being to reconstitute broad trends of land use over thousands of years, I am not sure its baseline map is as accurate as others that are available, such as for example the LUGE lab’s maps of cropland and pasture areas (available at http://www.earthstat.org/cropland-pasture-area-2000/), which is available at 5 minutes resolution. If the authors think that HYDE is the best choice of data in this case, perhaps a short justification would be in order. Also, the authors say the map is for 2017 but HYDE website says the current map is for 2000 and the HYDE 3.2.1. release note says that the maps go up to 2015. It would be good to clarify this (and confirm that it is indeed a map of cover and not a simulation that is being used in this analysis).

- Related to this, a more recent and possibly more accurate accessibility dataset might be worth considering: https://malariaatlas.org/research-project/accessibility_to_cities/ (published as doi:10.1038/nature25181). This also has a high resolution, of 1x1km2.

- Finally, the paper could explain better the selection and composition of the agricultural suitability index, considering the importance of that variable for the analysis. The authors say that “Global agricultural suitability is measured using an index reflecting the climatic, soil and topographical conditions necessary to grow the 16 most important food and energy crops” (l.153-155). The mention of 16 crops suggest that an index was built somehow from 16 suitability maps, begging the question of how. After looking up the source (https://zenodo.org/record/3748350) I realize that “The agricultural suitability represents for each pixel the maximum suitability value of the considered 16 plants”, but clarifying this in the paper would be better. As well, a few words about why this index is superior to others (e.g. the FAO GAEZ) would be welcome.

Spatial resolution: What is the rationale for the use of a 30 arc-min resolution? Unless I misunderstood, all datasets are available at a higher resolution (HYDE AND Verburg et al. at 5 arc-min, Zabel et al. at 30 arcsec), so I’m not sure why the analyses are run at 30 arc-min especially since it seems to cause some minor issues (l. 179 and following). Related to that, lines 181-2, it would be easier to understand if the granularity were expressed in terms of resolution (what do “two digits after the decimal point” represent?).

Theoretical crop distribution and crop composition: The authors use an index representing the max suitability for any crop in order to represent suitability. So, when mapping the potential crop distribution, we don’t actually know the composition of crops that would be possible in that distribution: maybe it is theoretically possible to cultivate the same area of crops in that area; but is it possible to cultivate the same crop mix? The lumping of all crops together masks the fact that reaching the potential distribution would likely imply a significant restructuring of agricultural production. I don’t think that is a fatal flaw but I think it warrants some discussion in the paper.

Implications of the results: I find that the authors jump a little too quickly to the conclusion that areas where the explanatory power of accessibility and suitability is low or allocation doesn’t seem to follow the best land are areas of suboptimal land allocation (lines 431 and following). It would be interesting to have a bit more discussion of what might explain that low explanatory power, first. For example, there may be political reasons that prevent the cultivation of some areas. Or, importantly, other uses that are not taken into account here (e.g., livestock herding, forestry) are competing with croplands. I think that the question of land competition is important to discuss here as cropland allocation doesn’t occur in a vacuum, so optimal cropland allocation may not mean optimal land allocation. Without such a contextualization, recommendations like those on lines 437-446 seem to come out of the blue.

Figure 1:

- In the left-hand side of the diagram, it wasn’t immediately clear to me that the divisions within each quantile were different parcels. While this becomes clear by reading the text, I think that adding a visual legend showing quantile, parcel, and cultivated areas (with arrows and text) would help make it more intuitive.

- I wonder if there would be a way to shade or hash that left part in such a way as to represent the theoretical distribution? That would provide another visual point of comparison. Just a suggestion, not a request.

- One alternative way to present that information would be having the cases arranged horizontally instead of vertically and aligning the four quantiles across different plots for each case. This would require compressing the left-hand plots a bit but might help readers link the plots mentally. Also just a suggestion.

- Finally, I wonder if the regression curves are useful here. That plot is not used in the rest of the paper, so it feels a bit like we’re putting effort into understanding a plot for nothing.

From Figure 1 and in the discussion referring to it, case 3 seems to reflect a situation in which there are important agglomeration economies – it is more interesting to develop agriculture close to where there is already some. I wonder if you could comment a bit on the role of agglomeration economies (…)

Figure 3: The shadings are a little confusing. I would suggest displaying both series of plots instead (accessibility and suitability).

Figure 4: Do the colours have a meaning? If not it might be good to either include these same colours in the theoretical example in figure 1, or to revert to black and white.

Figure 5: the labels are a bit cluttered, especially in panel A. Is there a way to avoid that the lines go over the labels?

Map: I think it would be good to have a zoomed-in map showing an example of how accessibility and suitability quantiles are distributed in a real-life territory. This would help grasp the concept better.

Minor remarks:

l.124: Typo: should be “a given parcel”.

l. 175 and throughout: I would suggest saying “Middle East and North Africa”, “Oceania and South-Eastern Asia”

l. 336-338: “However, in Eastern Europe, this positive effect can be at least partially offset by a strong negative and significant interaction between suitability and accessibility”: The plot indicates that India, Brazil, and RestAsia also have strong negative interactions. Why are they not included here?

l. 453: “market accessibility, which is sometimes overlooked in land-use studies”: that is a surprising statement to me, as it seems that accessibility is the one thing that most land use scholars all agree is important. There may be some studies that don’t take it into account but this suggests that it’s being ignored more systematically which I don’t think is accurate.

Reviewer #2: Dear Editors of PLOS ONE,

Thank you for the opportunity to review the manuscript ‘Assessing whether the best land is cultivated first: A quantile analysis.’ I find the effort to be an interesting and innovative approach to land use modeling and assessment, and I very much encourage it’s publication.

I did, however, have a few comments which, I think, might improve the draft.

First, the treatment of the economics of land use is very sparse. I do not feel that the authors need to delve too deeply into land use incentives, but I imagine that a greater depth of understanding of how rent theory has been used to frame recent land use discussions would benefit both the reader and the manuscript. I encourage the authors to dedicate some discussion to the use and application of von Thunen and ricardo in the recent literature.

Second, the graphics, and much of the nuanced reporting of various coefficients or tests, are difficult to interpret. Some of it may also not be necessary (see, for example, the reporting on Moran’s I- mentioned below). I encourage the authors to streamline their reported statistics, and revise their graphics for readability and interpretation.

Third, there is too little information on the data generating process. I remain unclear on how you determined market access or suitability. And there is no clarity on how accurate these measures are, how dynamic they are, or how error might be spatially correlated. I encourage you to offer more information on the data inputs. This could be addressed with limited text in the manuscript and more detailed information in the supplementary materials. Ideally, you might also provide your code and datasets. Maps of the spatial data would also be useful.

A few more minor comments follow below.

Abstract

‘Classical land rent theories imply that the best land is cultivated first.‘

This sentence could be re-written or dropped. Rent theory states that a location will be used for its most economically advantageous use. Generally speaking, this relates to the marginal returns to the use of labor and capital; while this usually translates to ‘the best land is used first’ this is not always the case. There are also tremendous questions here about what is meant by ‘best’. This article explicitly engages with this issue, but, if anything, it seems to highlight that the definition of best is hardly as clear as it is presented in this initial sentence.

Introduction

L52-53. Add citations for this sentence.

L79. I’m wondering how you change the suitability of land. How often does that happen? (more info is needed to understand how you classified suitability, etc.)

A map of agricultural suitability here would be very helpful.

L88-89 There is a decent case to make that regulations and ‘types of actors’ are elements of suitability; the former, especially, for land suitability.

Figure 1: the use of red as null squared at left, then as the histogram fill in the middle is confusing. In general, I find this graphic to be quite confusing.

Pages-6-7. I’m not sure I fully understand what the authors are trying to say here (as well as in figure 1). Perhaps the main point of these spatial schematics is this: ‘. Cropland can be preferentially distributed on the best land either by allocating crops to the least densely cultivated areas (better allocation) or by concentrating more crops on already densely cultivated areas (higher intensity) or both.’ But if so, I’m not following as to how density (given the article so far) should factor into the preferential distribution of agricultural land? Density is hardly a factor in classic rent theory (although some might say that this is a weakness…).

L150. ‘beginning of the 20th century…’ beginning? Or end?

L175. It would be useful here to see a map of suitability and access. It is surprising to see that 9% of EE is classified as unsuitable (where?). At the same time, one might assume that much of South America might be unsuitable, although in percent terms the relative quantity might be small.

L240. I’m not understanding why the authors are reporting that market access and suitability are spatially correlated. How could they be anything but?

L275. Why do the authors think that NA favors suitable land? Is it luck? Is it better data (on suitability)? Is it better transportation systems?

General: I think some might suggest that the relative importance of market access vs. suitability has evolved over time. Similarly, transport costs per unit of distance likely vary from place to place.

6. PLOS authors have the option to publish the peer review history of their article (what does this mean?). If published, this will include your full peer review and any attached files.

Reviewer #1: No

Reviewer #2: No

---

## [Author Response · Author response to Decision Letter 0]

4 Sep 2020

We would like to thank the two reviewers for their careful reading of our paper. Their remarks were of great help to us to improve the paper. You will find in the attached file our answers as well as the modifications made to the paper.

---

## [Decision Letter · Decision Letter 1]

5 Oct 2020

PONE-D-20-17711R1

Assessing whether the best land is cultivated first: A quantile analysis

PLOS ONE

Dear Dr. BRUNELLE,

Thank you for submitting your manuscript to PLOS ONE. After careful consideration, we feel that it has merit but does not fully meet PLOS ONE’s publication criteria as it currently stands. Therefore, we invite you to submit a revised version of the manuscript that addresses the points raised during the review process.

Please, pay attention to exceptions raised by reviewer 1 on the use of the word "best" as well as other words. You may consider providing an explicit definition of what the submission considers as "best" (e.g. most suitable and most accessible). I may have accessed the wrong document, but reference 16 did not use the term "best". However, you may choose to interpret "most fertile and most favorable" as best, and provide this definition well ahead in the introduction to avoid any doubt. Kindly pay attention to the other two major concerns, and address them or provide  clear reasons why they need not be addressed.

We look forward to receiving your revised manuscript.

Kind regards,

Gerald Forkuor

Academic Editor

PLOS ONE

Reviewers' comments:

Reviewer's Responses to Questions

**Comments to the Author**

1. If the authors have adequately addressed your comments raised in a previous round of review and you feel that this manuscript is now acceptable for publication, you may indicate that here to bypass the “Comments to the Author” section, enter your conflict of interest statement in the “Confidential to Editor” section, and submit your "Accept" recommendation.

Reviewer #1: All comments have been addressed

Reviewer #2: (No Response)

2. Is the manuscript technically sound, and do the data support the conclusions?

Reviewer #1: Yes

Reviewer #2: Yes

3. Has the statistical analysis been performed appropriately and rigorously? 

Reviewer #1: Yes

Reviewer #2: Yes

4. Have the authors made all data underlying the findings in their manuscript fully available?

Reviewer #1: Yes

Reviewer #2: Yes

5. Is the manuscript presented in an intelligible fashion and written in standard English?

Reviewer #1: Yes

Reviewer #2: Yes

6. Review Comments to the Author

Reviewer #1: I would like to thank the authors for addressing my comments. I am looking forward to seeing this paper in print.

Reviewer #2: Dear PLOS One,

Thank you very much for the opportunity to re-review the research article ‘Assessing whether the best land is cultivated first: A quantile analysis’. I have included a number of small suggestions below which I think might improve the manuscript.

I ask the editor to please make sure that the authors address the three concerns that I addressed in my initial review but which were not addressed in this revision:

1. The use of the term ‘best’ and the passive use of ‘allocation’, both of which are gross oversimplifications of, or even contrary. to rent theory)

2. The discussion of moran’s i in the methods (not needed here)

3. Please add maps of biophysical and access suitability to the main document.

Once the authors address these concerns and, ideally, address the points below, I recommend this manuscript for publication.

Thank you so much.

Abstract:

1st sentence. I continue to find this sentence as overly-simplistic and awkward. Please (please!) re-write this sentence to better align with classical rent theory or this current analysis. I would avoid using words like ‘best’ and ‘first’, since I’m not sure what constitutes ‘the best’ land or being used ‘first’. I raised my concern with this sentence in the first draft and continue to raise it here.

Introduction.

L39. I would hesitate to refer to sources 8-9 as foresight studies. I think that ‘projections’ or something similar is much better word choice.

L53. The wording ‘is allocated’ should be changed. This suggests that someone is doing the allocation, so to speak. This is presumably true for individual properties or land portfolios. But I don’t think that there are many scenarios where, outside of basic zoning laws, farmland use is allocated by land use planning commissions. Change the wording here to refer to influence rather than allocations. The former implies local decisions, whereas the latter implies some sort of authoritarian decision process. This use of allocated recurs throughout. Please adjust.

L60 ‘Many global land use models are rooted in classical rent theories by allocating land according to a profit function that depends on the intrinsic qualities of land provided by vegetation models (usually in terms of climatic potential yields) or based on index of agricultural suitability.’

Global land use models are generally rooted in elasticities and responses to market possibilities or climate trends. Classical rent theory, of course, is implicit in these relationships.

L67.

‘However, at the local scale, several empirical studies show inconsistent responses to agricultural suitability and market accessibility across regions24. Although strong relationships have been observed between cultivated area and agricultural suitability and market accessibility in several European regions, some croplands in China were recently moved to less fertile areas in response to urbanization dynamics.

This is very much aligned with classical rent theory, not inconsistent with it.

L69. Moreover, assessments carried out at the land system scale show contradictory effects of spatial determinants on land allocation24.

I’m not sure what this sentence is trying to say, why it is included, or what effects that are referring to.

L101. The use of the term ‘best land’ continues to be problematic. Do they mean ‘optimal’ in terms of suitability and access?

L258. This paragraph can be deleted or moved to the SI. I’m not sure what it adds. (somewhat annoyingly, I raised this clearly in the initial review)

L455. The note on fits by region is quite interesting and begs more discussion. Presumably, it’s not only that land optimization has increased over time, but that transportation costs have also shifted with time.

L496. This discussion of ‘grand theories’ and the warning against oversimplification in their application is in many respects and oversimplification in the characterization of these theories. In effect, there is tremendous nuance in how the theory is applied and manifested in land economics. While I find the simplicity of this analysis to be useful for clarifying a widely used model, the authors at times show a simplicity in their knowledge of the field and how these theories are used.

Map figure. Please include a map of the suitability and access regions in the main text.

7. PLOS authors have the option to publish the peer review history of their article (what does this mean?). If published, this will include your full peer review and any attached files.

Reviewer #1: No

Reviewer #2: No

---

## [Author Response · Author response to Decision Letter 1]

22 Oct 2020

L39. I would hesitate to refer to sources 8-9 as foresight studies. I think that ‘projections’ or something similar is much better word choice. 

Authors'response: done

L53. The wording ‘is allocated’ should be changed. This suggests that someone is doing the allocation, so to speak. This is presumably true for individual properties or land portfolios. But I don’t think that there are many scenarios where, outside of basic zoning laws, farmland use is allocated by land use planning commissions. Change the wording here to refer to influence rather than allocations. The former implies local decisions, whereas the latter implies some sort of authoritarian decision process. This use of allocated recurs throughout. Please adjust. 

Authors'response: We agree that the word "allocated" implies that someone makes the allocation, but this does not imply anything about the type of decision making. An individual can allocate resources without any form of authoritian process. Nevertheless, in order to avoid any ambiguity, we have replaced the term "is allocated" by "is used".

L60 ‘Many global land use models are rooted in classical rent theories by allocating land according to a profit function that depends on the intrinsic qualities of land provided by vegetation models (usually in terms of climatic potential yields) or based on index of agricultural suitability.’ Global land use models are generally rooted in elasticities and responses to market possibilities or climate trends. Classical rent theory, of course, is implicit in these relationships. 

Authors'response: True. We add in the paper the following comment (l.64): "Land supply elasticities are also generally used to determine land conversion rates in a given location. In this case, the elasticity is estimated based on assumptions derived from rent theories (Villoria et al., 2018)". 

L.67 ‘However, at the local scale, several empirical studies show inconsistent responses to agricultural suitability and market accessibility across regions24. Although strong relationships have been observed between cultivated area and agricultural suitability and market accessibility in several European regions, some croplands in China were recently moved to less fertile areas in response to urbanization dynamics. This is very much aligned with classical rent theory, not inconsistent with it. 

Authors'response: True. We removed the word "inconsistent" and we have modified the sentence l.68 as follows: "Several empirical studies at the local scale have investigated responses to agricultural suitability and market accessibility".

L69. Moreover, assessments carried out at the land system scale show contradictory effects of spatial determinants on land allocation24. I’m not sure what this sentence is trying to 

say, why it is included, or what effects that are referring to. 

Authors'response: For more clarity, we have modified the sentence as follows (l.72-74): "At the land system level, several assessments reported ambiguous effects of some spatial determinants on land use, like for example a negative effect of market accessibility on agricultural land use. 25". 

L101. The use of the term ‘best land’ continues to be problematic. Do they mean ‘optimal’ in terms of suitability and access? 

Authors'response: We have clarified this point l. 59-60 that: "In this paper, we refer to the highest grades of land in terms of potential productivity, location suitability or both as “best land". And in lines 80-81: "In this paper, we refer to the areas with the highest suitability and/or accessibility indices as "best land".

L258. This paragraph can be deleted or moved to the SI. I’m not sure what it adds. (somewhat annoyingly, I raised this clearly in the initial review) 

Authors'response: We agree that the Moran test does not bring much information. It is only a confirmation. That is why we have replaced the term "reveals" by "confirms". However, we do not wish to remove this paragraph as we think it is important to include an explanation on how spatial auto-correlation has been treated in the paper. 

L455. The note on fits by region is quite interesting and begs more discussion. Presumably, it’s not only that land optimization has increased over time, but that transportation costs have also shifted with time. 

Authors'response: Reduction in transportation cost is indeed a key driver of the optimization process. We clarify this point in the discussion l.475-478: "The reduction in transport costs resulting from the development of the railways, combined with the reduction in transatlantic freight rates, made it possible to cultivate the fertile lands of the Midwest and export part of the agricultural production to Western Europe."

L496. This discussion of ‘grand theories’ and the warning against oversimplification in their application is in many respects and oversimplification in the characterization of these theories. In effect, there is tremendous nuance in how the theory is applied and manifested in land economics. While I find the simplicity of this analysis to be useful for clarifying a widely used model, the authors at times show a simplicity in their knowledge of the field and how these theories are used. 

Authors'response: We agree that the theory can be applied in different ways. Here the term "grand theory" is not pejorative. It refers to the distinction made by Meyfroidt et al. between middle-range theory and general theoretical frameworks. This distinction is very nuanced as it accepts the need for generalization. The expression "oversimplified conception" is not ours and refers to the Boserupian conception of the use of cultivated land on which we are unable to conclude. 

Map figure. Please include a map of the suitability and access regions in the main text. 

Authors'response: done

---

## [Editor Report · Decision Letter 2]

29 Oct 2020

Assessing whether the best land is cultivated first: A quantile analysis

PONE-D-20-17711R2

Dear Dr. BRUNELLE,

We’re pleased to inform you that your manuscript has been judged scientifically suitable for publication and will be formally accepted for publication once it meets all outstanding technical requirements.

Kind regards,

Gerald Forkuor

Academic Editor

PLOS ONE
---

## [Editor Report · Acceptance letter]

1 Dec 2020

PONE-D-20-17711R2 

Assessing whether the best land is cultivated first: A quantile analysis 

Dear Dr. Brunelle:

I'm pleased to inform you that your manuscript has been deemed suitable for publication in PLOS ONE. Congratulations! Your manuscript is now with our production department. 

Kind regards, 

on behalf of

Dr. Gerald Forkuor 

Academic Editor

PLOS ONE